# Investigation of the Expression Pattern and Functional Role of miR-10b in Intestinal Inflammation

**DOI:** 10.3390/ani13071236

**Published:** 2023-04-02

**Authors:** Zijuan Wu, Guolin Pi, Wenxin Song, Yali Li

**Affiliations:** 1Hunan International Joint Laboratory of Animal Intestinal Ecology and Health, Hunan Normal University, Changsha 410081, China; 2Hunan Provincial Key Laboratory of Animal Intestinal Function and Regulation, Hunan Normal University, Changsha 410081, China

**Keywords:** miR-10b, weaning, piglet, colitis, inflammatory response

## Abstract

**Simple Summary:**

MicroRNAs (miRNAs) play a vital role in maintaining the homeostasis of the intestinal immune system, and dysregulation of miRNAs expression has been linked to several inflammatory diseases. Although miRNAs possess remarkable potential as molecular markers for monitoring diseases and disorders, their implications for animal health and welfare in livestock remain elusive. In the swine industry, weaning is the most challenging period for piglets, associated with a peak of intestinal immune activation and intestinal epithelium injury. Thus, it is interesting to identify suitable miRNAs as biomarkers for monitoring weaning-related disorders during the suckling-to-weaning transition in piglets and to investigate the potential of them being used as anti-inflammatory strategies for alleviating stress. This study has revealed a distinctive miR-10b expression pattern in the intestinal tissue of piglets following weaning and a critical role for miR-10b in immune regulation using a murine colitis model. These findings may provide new approaches for potential future interventions for swine health management.

**Abstract:**

Implications of miRNAs for animal health management in livestock remain elusive. To identify suitable miRNAs as monitoring biomarkers, piglets were randomly selected for sampling on days 0, 1, 3, 7, and 14 post-weaning. The results show that miR-10b levels in the villus upper cells of the jejunum on days 3 and 7 were significantly lower than that on day 14 post-weaning and reduced by approximately 30% on day 3 and 55% on day 7 compared to day 0. In contrast, miR-10b in crypt cells decreased by approximately 82% on day 7 and 64% on day 14 compared with day 0. Next, miR-10 knockout mice and wild-type mice were subjected to dextran sulfate sodium (DSS) for 7 days. The findings demonstrate that mice lacking miR-10b were more susceptible to DSS administration, as demonstrated by worse survival, greater weight loss, more severe tissue damage, and increased intestinal permeability. Moreover, the increased disease severity was correlated with enhanced macrophage infiltration, coincident with significantly elevated pro-inflammatory mediators and immunoglobulins. Bioinformatic analysis further reveals that the enriched pathways were mainly involved in host immune responses, and Igtp was identified as a potential target of miR-10b. These findings may provide new strategies for future interventions for swine health and production.

## 1. Introduction

MicroRNAs (miRNAs), a class of small non-coding RNA molecules with 19–24 nucleotides in length, play a vital role in post-transcriptional gene silencing by targeting the 3′ untranslated region (UTR) of target genes [1]. Evidence has shown that miRNAs are involved in almost every biological process, including cell proliferation, differentiation, apoptosis, and stress response [2]. Moreover, miRNAs have emerged as key gene regulators to control inflammation and prevent the uncontrolled progress of inflammatory reactions, playing a central role in maintaining the homeostasis of the immune system [3]. The dysregulation of miRNA expression has been linked to a wide variety of inflammatory diseases, such as asthma, chronic obstructive pulmonary disease, and inflammatory bowel disease [4,5,6]. Several miRNAs, such as miR-31, miR-155, and miR-223, have been recognized as novel biomarkers for the early diagnosis and prognosis of colitis as well as new targets for molecular treatment [7,8,9]. While miRNAs possess remarkable potential as molecular markers for monitoring diseases and disorders, their implications for animal health management in livestock remain elusive.

Owing to high sequence homology across species, miRNAs can be readily detected without the need for species-specific assays. Accordingly, the identification of miRNAs as biomarkers for monitoring stress-related disorders in livestock is an interesting field of research, which may provide an in-depth overview of animal welfare and related health conditions at the molecular level [10]. In the swine industry, weaning is the most challenging period for piglets, associated with a peak of intestinal immune activation and intestinal epithelium injury [11]. Weaning also disturbs the crypt–villus differentiation, resulting in intestinal villus atrophy and crypt hyperplasia in piglets. A previous study has shown a distinctive miRNA expression profile in the small intestine between weaning and suckling piglets, and the difference varied with the number of days after weaning [12]. However, the effects and mechanism of differentially expressed miRNAs on intestinal damage during weaning stress have not been elucidated. As a member of the miRNA family, miR-10b (miR-10b-5p) has been reported to play a vital role in the regulation of inflammation in gastrointestinal motility disorders [13] and to be upregulated in the villus upper cells of the jejunum of piglets compared to crypt cells, indicating a functional role in porcine intestines [14]. In this study, to assess the capability of miR-10b to be used as a biomarker associated with weaning, the dynamic expression pattern of miR-10b in the porcine intestinal mucosa was examined on days 0, 1, 3, 7, and 14 after weaning. Loss-of-function study in vivo has been shown as an efficient approach for better understanding the functional role of miRNAs [15]. Given the high dosage of antagomirs required to completely silence miR-10b for piglets, genetically engineered mice deficient in miR-10b were used as alternative models. In the current study, miR-10 knockout (KO) mice and wild-type (WT) mice were subjected to experimental DSS-induced colitis, a well-established animal model of mucosal inflammation, to further clarify the contribution of miR-10b to intestinal inflammation and homeostasis.

## 2. Materials and Methods

### 2.1. Isolation of Intestinal Cells from Weaning Piglets

Twenty piglets (Duroc × (Landrace × Large Yorkshire), average weight (6.53 ± 0.14) kg) were weaned at 21 days old. Diets were formulated to meet the standard nutrient requirements for weaned piglets, and all piglets had free access to feed and water during the experiment. The experiment duration lasted 14 days. Four piglets per time point were randomly sacrificed on days 0, 1, 3, 7, and 14 after weaning for sampling. The sequential isolation of piglet jejunum epithelial cells along the crypt–villus axis was performed according to a previously described method [14]. Briefly, the divided mid-jejunum segments were washed and incubated at 37 °C for 30 min with oxygenated (O_2_/CO_2_, 19:1) PBS buffer. Then, the jejunum segments were incubated in oxygenated EDTA chelating buffer at 37 °C for 40 min. The chelating buffers were then centrifuged at 1000× rpm for 10 min at 4 °C. This procedure was repeated three times to yield three “cell fractions”. Cells collected the first time were the intestinal villus upper cells, and at the third time were crypt cells. These collected cells were washed twice with oxygenated suspension buffer, centrifuged at 1000× rpm for 10 min, rapidly frozen in liquid nitrogen, and stored at −80 °C for further analysis.

### 2.2. Quantitative Real-Time PCR Analysis

Total RNA was extracted from mouse colon tissues using TRIzol reagent (Takara, Dalian, China) according to the manufacturer’s protocol. RNA samples were reverse transcribed to cDNA using PrimeScript RT Master Mix kit (Takara, Dalian, China). The cDNA samples were then tested for the expression of Igtp (interferon gamma-induced GTPase), and results were normalized to β-actin expression. To measure the expression levels of miR-10b, small RNAs were extracted from piglet intestinal epithelial cells or mouse colon tissues using RNAiso for small RNA kit (Takara) and reverse transcribed using the Mir-X miRNA First-Strand Synthesis Kit (Takara, Dalian, China). The level of miR-10b was normalized to U6 (small nuclear RNA) expression. Real-time quantitative PCR was performed using Quanstudio 5 (Thermo Fisher, Singapore) using SYBR Green qPCR Master Mix (Thermo Fisher, MA, USA). Each PCR reaction was performed in triplicate. Relative quantification was calculated using the 2^−ΔΔCT^ method. The primers used in this study are listed in Table 1.

### 2.3. DSS-Induced Colitis in Mice

Six-to-eight-week-old female C57BL/6 (wild-type, WT) mice were obtained from the Hunan Silaike Jingda Laboratory Animal Co., Ltd., (Changsha, China), and miR-10b knockout (KO) mice (generated on C57BL/6 background) were purchased from Nanjing Biomedical Research Institute of Nanjing University (Nanjing, China). All mice were fed autoclaved food and water and maintained in a specific-pathogen-free facility at Hunan Normal University (Changsha, China). To induce colitis, WT and miR-10b KO mice were given drinking water containing 4% DSS (MW 36,000–50,000; Meilunbio, Dalian, China) for 7 consecutive days. Mice in control groups were given normal drinking water only. Survival rates, body weight loss, rectal bleeding, and stool consistency were monitored daily. The disease activity index (DAI) scores were the sum of body weight loss, fecal hardness, and bleeding scores as previously described (Appendix A) [16]. After 7 days of DSS treatment, mice were euthanized by CO_2_ inhalation. Serum and colon samples were harvested and stored in a refrigerator at −80 °C for subsequent experiments.

### 2.4. Histopathological Analysis

Colonic tissues were removed, fixed in 10% formalin, and embedded in paraffin. Then, 4-μm sections were cut and stained with hematoxylin and eosin (H&E). Images were obtained under a microscope (Leica DM3000; Wetzlar, Germany). Scores were assessed by determination of infiltration of inflammatory cells (score range, 0–4), together with the evaluation of cecal tissue damage (score range, 0–4), as described previously (Appendix A) [17]. Stained sections were analyzed without prior knowledge of the treatment.

### 2.5. FITC-Dextran Intestinal Permeability Assay

Mice were fasted with water and food for 4 h, followed by oral gavage of fluorescein isothiocyanate (FITC)-dextran (60 mg per 100 g bodyweight; Sigma-Aldrich, St. Louis, MO, USA). Blood samples were collected 4 h post-gavage and centrifuged at 3000 rpm for 10 min at 4 °C. FITC-dextran signal was determined by fluorescence microplate detector (490/525 nm).

### 2.6. Immunofluorescence Microscopy

Paraffin-embedded colon samples were sectioned. After dewaxing and hydration, antigen retrieval was achieved by immersing the sections in a citrate buffer. Slides were then blocked with avidin/biotin agent (Vector Laboratories, Burlingame, CA, USA). To analyze the infiltration of macrophages and neutrophils, colonic slides were stained with FITC anti-mouse F4/80 (a murine macrophage-restricted cell surface glycoprotein) (eBioscience, San Diego, CA, USA) and PE/Cyanine5-labeled anti-mouse Ly6G (lymphocyte antigen 6 complex locus G) antibodies (Biolegend, San Diego, CA, USA). The nuclei were counterstained with DAPI (Vector Laboratories). Slices were sealed with anti-fluorescence quenching solution and analyzed under a fluorescence microscope (Leica DM3000; Wetzlar, Germany). The mean number of F4/80^+^ cells detected in each high-power field was calculated by counting four fields from each sample (samples from three mice per group were counted).

### 2.7. Enzyme-Linked Immunosorbent Assay

Serum levels of lipopolysaccharide (LPS), diamine oxidase (DAO), D-lactate (D-LA), tumor necrosis factor α (TNF-α), interleukin-1β (IL-1β), interferon-γ (IFN-γ), immunoglobulin M (IgM), IgA, and IgG were measured with commercially available ELISA kits (eBioscience, CA, USA) according to the manufacturer’s instructions.

### 2.8. Bioinformatic Analysis

After DSS treatment, total RNA of colon tissues was extracted using TRIzol reagents according to instructions (*n* = 4/group). High-quality RNA (RIN value > 9.0) was used for subsequent RNA sequencing analysis using the Illumina Hiseq platform using 2 × 150 bp paired-end (PE) sequencing (Majorbio Bio-pharm Technology Co., Ltd., Shanghai, China). RNA sequencing libraries were constructed with Illumina TruseqTM RNA sample prep Kit. Approximately 49.72 and 51.24 million raw reads were obtained from DSS-treated miR-10b KO and WT mice, respectively (Appendix A). After checks for read quality with FASTP, 49.20 and 50.72 million high-quality clean reads remained for further analysis. The Q30 values were over 93.29%, and the GC content varied from 48.69 to 50.27%. Clean reads were mapped to the mouse reference genome (GRCm39) using HISAT2 software, and the mapping ratios for DSS-treated miR-10b KO and WT mice were 86.05 and 87.67%, respectively (Appendix A). Differential expression analysis was performed using the DESeq2 package, and genes with |log2FC| ≥ 1 and an adjusted *p*-value of <0.05 were considered to be significantly different expressed genes (DEGs). Metascape (https://metascape. org; accessed on 1st December, 2022), a web-based portal, was applied to perform functional enrichment analysis [18]. Terms with a *p*-value < 0.01, a minimum count of 3, and an enrichment factor >1.5 were grouped into clusters based on their membership similarities. *p*-values were calculated based on the cumulative hypergeometric distribution, and Q-values were calculated using the Benjamini–Hochberg procedure (Metascape, v3.5.20230101). Gene ontology (GO) analysis was used to expound promising signaling pathways correlated with DEGs. Kyoto Encyclopedia of Genes and Genomes (KEGG) analysis was carried out by KOBAS, with Bonferroni-corrected *p*-value < 0.05. Transcription Regulatory Relationships Unraveled Sentence-based Text mining (TRRUST) database was carried out to predict the core regulatory transcription factors. Target genes of miR-10b were predicted using StarBase database (https://starbase.sysu.edu.cn/starbase2/index.php; accessed on 1 December 2022) [19].

### 2.9. Dual Luciferase Activity Assays

To generate reporter constructs for luciferase assays, 315 bp fragments containing predicted miR-10b target site in the 3′ UTR of Igtp and its identical sequence with mutation, were cloned into the pmiR-RB-REPORT™ luciferase reporter vector (RiboBio, Guangzhou, China) between the XhoI and NotI sites. The miR-10b mimics and negative controls were designed and synthesized by RiboBio (Guangzhou, China). HEK293T cells were cultured under standard conditions and co-transfected with the vectors, mimic-miR-10b, or negative controls using Lipofectamine 2000 (Invitrogen, Carlsbad, CA, USA). At 48 h after transfection, relative luciferase activity was consecutively measured with the Dual Luciferase Reporter Assay Kit according to the manufacturer’s instructions (Promega, Madison, WI, USA). Results were expressed as the intensity ratio of Renilla to Firefly luciferase. Experiments were assayed in triplicate and repeated twice.

### 2.10. Statistical Analysis

Data are shown as the mean ± SEM. Survival curves of mice with DSS treatment were compared using Kaplan–Meier analysis and log-rank test. Statistical differences were determined using unpaired Student’s t test or one-way ANOVA with Tukey’s post hoc test using GraphPad Prism (GraphPad Software, San Diego, CA, USA). Enrichment analysis was examined by Metascape utilizing the hypergeometric test and Benjamini–Hochberg *p*-value correction algorithm. Statistical significance was defined as a *p*-value < 0.05.

## 3. Results

### 3.1. Dramatic Changes in miR-10b Expression Levels in Piglets following Weaning

The expression levels of endogenous miR-10b in the villus upper cells of the jejunum in weaning piglets on days 3 and 7 were significantly lower than that of day 14 post-weaning (*p* < 0.05). Moreover, the levels of miR-10b in villus upper cells reduced by approximately 30% on day 3 and 55% on day 7 compared to day 0, while there was no significant difference (Figure 1A). In contrast, the highest expression level of miR-10b in crypt cells was observed on day 1 when compared with day 0 (*p* < 0.05). Although it did not reach significance, the levels of miR-10b in crypt cells decreased by approximately 82% on day 7 and 64% on day 14 when compared to day 0 (Figure 1B). Additionally, it is worth noting that miR-10b was up-regulated in villus upper cells compared to crypt cells on days 7 and 14, with the greatest ratio than other time points post-weaning (*p* < 0.05) (Figure 1C).

### 3.2. miR-10b Deficiency Increased Susceptibility to DSS-Induced Colitis

As shown in Figure 2A, miR-10b was significantly down-regulated in the colonic tissues of WT mice with DSS treatment (*p* < 0.01). Next, a knockout mouse model was used to further study the physiological role of miR-10b. Compared with WT animals, miR-10b KO mice had much more severe colitis as indicated by worse survival (Figure 2B), greater weight loss (Figure 2C), an increased disease activity index (Figure 2D), and significant colon shortening (Figure 2E,F) after DSS administration (*p* < 0.05). Moreover, significantly worse colon inflammation was observed by HE staining in miR-10b KO mice (Figure 2G,H), evidenced by more severe epithelial erosion, marked crypt damage, and increased inflammatory cell infiltration (*p* < 0.05). Consistent with these findings, significantly increased intestinal permeability was observed in miR-10b KO mice after DSS treatment (*p* < 0.01), suggesting impaired barrier function in the colonic epithelium of miR-10b KO mice.

### 3.3. miR-10b Deficiency Exacerbated Immune Response after DSS Treatment

The inflammatory infiltrating cell population was characterized using an immunofluorescence approach. As shown in Figure 3A, a clear infiltration of F4/80+ macrophages and Ly6G+ neutrophils was observed in the colon tissues of mice after DSS administration. Notably, increased macrophage numbers were found in the colonic lamina propria of miR-10b KO mice (Appendix A). Consistently, elevated serum levels of LPS, DAO, D-LA, TNF-α, IL-1β, IFN-γ, IgM, IgA, and IgG were observed in mice with DSS treatment in contrast to control mice receiving normal drinking water (Figure 3B–J). Furthermore, loss of miR-10b significantly increased serum levels of LPS, DAO, D-LA, TNF-α, IL-1β, IFN-γ, IgM, and IgA after DSS treatment compared with WT mice (*p* < 0.01). In addition, even without DSS treatment, the ablation of miR-10b also elevated concentrations of LPS, TNF-α, IL-1β, and IFN-γ in the serum of mice when compared to WT mice (*p* < 0.01).

### 3.4. Bioinformatic Analysis of Key Pathways in miR-10b KO Mice after DSS Treatment

A total of 507 differentially expressed genes (DEGs), including 241 up-regulated and 266 down-regulated genes, between miR-10b KO mice and WT mice with DSS treatment, were demonstrated in Figure 4A. Using GO analysis, the top 20 significantly enriched GO terms were identified, and the most enriched ones were associated with innate immune response, defense response, and leukocyte migration (Figure 4B). KEGG analysis revealed that complement and coagulation cascades, rheumatoid arthritis, IL-17 signaling pathway, cell adhesion molecules, cytokine–cytokine receptor interaction, antigen processing, and presentation were enriched significantly (Figure 4C). Using TRRUST enrichment analysis, the top 20 core regulatory transcription factors were predicted, and most of the DEGs were found to be regulated by transcription factor Jun, Nfkb1, Rfx5, Rfxank, and Rfxap (Figure 4D).

### 3.5. Identification of Igtp as a Target of miR-10b

A total of 16 overlapped genes, between 1530 predicted target genes of miR-10b and 542 significantly up-regulated genes in miR-10b KO mice with DSS treatment, are shown in Figure 5A. Igtp, one of the overlapped genes, was considered a potential key target gene of miR-10b. The putative miR-10b binding site in the 3′ UTR of Igtp was demonstrated in Figure 5A. Moreover, the up-regulated level of Igtp in miR-10 KO mice after DSS administration was further confirmed using RT-PCR (Figure 5B). Next, the ability of miR-10b to regulate the 3′ UTR of Igtp was evaluated via luciferase reporter assays. As illustrated in Figure 5C, co-transfection with miR-10b mimics suppressed the activity of luciferase activity of the vector with the wild-type Igtp 3′ UTR (*p* < 0.01). Moreover, the mutation of the miR-10b recognition site abrogated the repressive ability, demonstrating the specificity of the target sequence for Igtp (Figure 5C).

## 4. Discussion

Weaning induces dramatic changes in intestinal development and barrier function, as well as mucosal immunity and homeostasis [11,20,21]. However, profiles of miRNA expression in the porcine intestine during early weaning are relatively unknown. Here, we demonstrated a distinct expression pattern of miR-10b in intestines after weaning. The miR-10b levels in villus upper cells on days 3 and 7 were significantly lower than that on day 14 post-weaning and strikingly reduced compared to day 0. These findings are consistent with the Jang and Lee [22] study, which performed microRNA expression profiling during the suckling-to-weaning transition in piglets to identify weaning-associated differentially expressed miRNAs. In that research, the authors found that miR-10a-5p, another member of the highly conserved miR-10 family, had a decreased expression at 1 week after weaning and a stable expression at 2 weeks after weaning. This expression pattern of miR-10a-5p was similar to that of miR-10b observed in the villus upper cells in our study. Furthermore, we found that miR-10b expressed differently along the villus–crypt axis. The expression of miR-10b was up-regulated in villus upper cells compared to crypt cells on days 7 and 14 post-weaning in our study, which agrees with a previous study reporting that miR-10b was up-regulated in the villus upper cells compared to crypt cells of piglets at 21 days old [14]. Evidence has shown that the most serious effects of weaning on intestines took place in the first week after weaning [12]. In addition, another report established patterns of inflammation markers in the time subsequent to weaning and revealed a post-weaning mucosal inflammation response for at least 15 days post-weaning [23]. Given that miRNAs have been reported to modulate inflammation [5,24], the drastically reduced miR-10b expression might be involved in the dysregulated immune reaction caused by weaning stress.

Next, the role of miR-10b in inflammation-associated intestines was further determined. Consistently, miR-10b was significantly down-regulated in response to DSS treatment, which agrees with previous research showing reduced miR-10b expression after DSS insults [16]. Notably, defects in miR-10b resulted in increased morbidity and mortality, evidenced by worse survival, greater weight loss, significant colon shortening, more severe tissue damage, and increased intestinal permeability. Disruption of the intestinal barrier integrity was further confirmed by the elevated serum level of permeability-related parameters, such as LPS, DAO, and D-LA. This result is in accordance with a previous study, showing that miR-10b KO mice were more sensitive to DSS-induced colitis as indicated by severe disruption of colonic barrier function [16]. Similarly, miR-148a KO mice have been shown to exhibit severe colitis characterized by greater weight loss, worse survival, and increased colon shortening, while restoring miR-148a expression alleviated the severity of inflammation [25]. Damaged intestinal barrier function was also found in mice lacking DICER1, the obligatory miRNA-processing enzyme, thereby resulting in aggravated intestinal inflammation [26].

Moreover, increased disease severity was correlated with exaggerated macrophage recruitment, coincident with significantly elevated pro-inflammatory mediators and immunoglobulins. Recently, a previous study found that miR-31 KO mice developed more severe colitis as evidenced by increased numbers of leukocytes and macrophages, elevated levels of pro-inflammatory cytokines, as well as decreased levels of IL-10 after DSS treatment [27]. Consistent with our findings, another study has demonstrated that serum inflammatory factor levels were potently increased in miR-10b KO mice, indicating that the loss of miR-10b may result in aggravated intestinal and systemic inflammation in DSS-induced mice [16]. Collectively, these findings indicated that miR-10b deficiency significantly exacerbated the immune response after DSS treatment. Further investigation is needed to determine whether restoring miR-10b abundance could attenuate inflammation induced by DSS.

Integrated bioinformatic analysis was applied to further identify key modules and pathways involved in the development of colitis in miR-10b KO mice. Enrichment for GO and KEGG pathway analysis revealed that these obtained DEGs were mainly associated with innate immune response, defense response, leukocyte migration, complement and coagulation cascades, IL-17signaling pathway, cytokine–cytokine receptor interaction, and antigen processing and presentation, closely related to host immunity. This result was in line with the hyper-reactive immune reactions observed for miR-10b KO mice in the present study. Meanwhile, according to the TRRUST analysis, most of the DEGs were found to be regulated by the transcription factors Jun, Nfkb1, Rfx5, Rfxank, and Rfxap. NF-κB is a well-known key regulator of inducible gene expression in the immune system, and its activation is markedly induced in colitis patients [28,29]. A previous study also demonstrated that NF-κB signaling pathways were over-activated in miR-31 KO mice in response to DSS treatment [27]. Together with our current work, these findings indicate that NF-κB signaling is critical for driving colitis and the hyperinflammatory conditions observed in miR-10b KO mice.

Igtp was first identified as a novel inducibly expressed GTPase regulated by IFN-γ in macrophages [30], encoding 48-kDa GTP-binding proteins that were localized to the endoplasmic reticulum [31]. Igtp-deficient mice were reported to display a profound loss of host resistance to acute infections of parasites, suggesting that Igtp defines a potential IFN-γ- mediated pathway with a specialized role in antimicrobial resistance [31]. In another study, it was found that IGTP was necessary for toxoplasma vacuolar disruption and induced parasite egression in IFN-γ-stimulated astrocytes [32]. However, the regulatory role of Igtp in intestinal mucosal inflammation was relatively unknown. In the current study, Igtp was significantly up-regulated in miR-10b KO mice after DSS treatment. Using a dual luciferase assay, Igtp was confirmed as a target gene for miR-10b. Therefore, it may indicate that the association between miR-10b and Igtp signaling pathways may be involved in the pathogenesis of experimental colitis. Given the high levels of IFN-γ observed in miR-10b KO mice, the enhanced expression of Igtp may result from the deficiency of miR-10b or the elevated level of IFN-γ in serum, which needs further research. Based on the findings reported here, it will be necessary for future work to examine the potential role of Igtp in immune modulation and inflammation.

## 5. Conclusions

In summary, this study has revealed a distinctive miR-10b expression pattern in the intestinal tissues of piglets following weaning and a critical role for miR-10b in immune regulation using a murine colitis model. Functional enrichment analysis further described the potential functions of miR-10b. Moreover, Igtp was identified as a potential target of miR-10b using a dual luciferase assay. Further studies using miRNA mimetics or other modified miRNAs may aid in determining their potential utility as novel anti-inflammatory agents to attenuate weaning stress. These findings may also provide new approaches for future interventions for swine health management.

## Figures and Tables

**Figure 1 animals-13-01236-f001:**
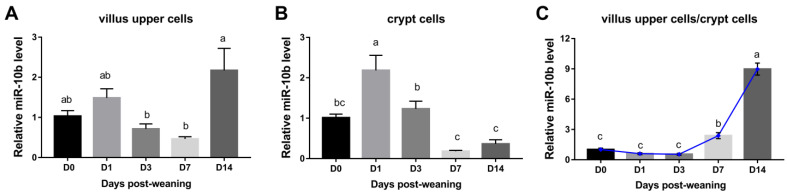
Changes in miR-10b expression levels in piglets during the post-weaning period. Expression of endogenous miR-10b in (**A**) villus upper cells, (**B**) crypt cells, and (**C**) the ratio of villus upper cells to crypt cells of jejunum on days 0, 1, 3, 7, and 14 post-weaning (*n* = 4/group). Labeled without a common superscript (alphabetic letters) means differ significantly, *p* < 0.05.

**Figure 2 animals-13-01236-f002:**
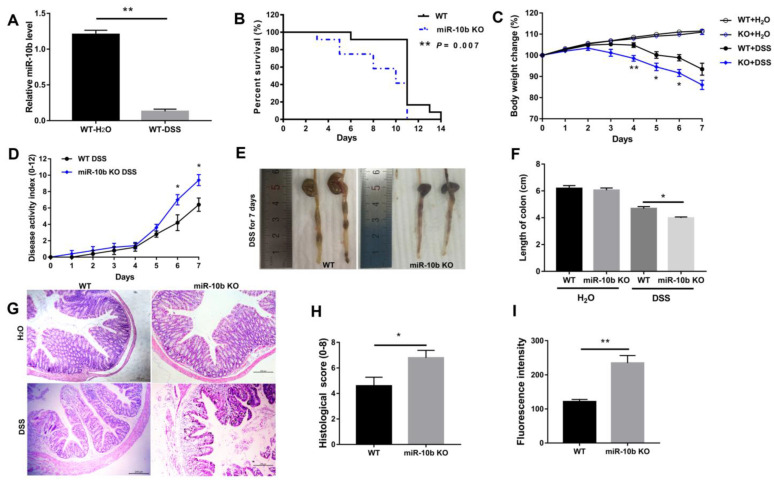
miR-10b KO mice were more susceptible to DSS-induced colitis. (**A**) Changes in miR-10b expression in colon tissues from WT mice following 7 days of DSS treatment (*n* = 6/group; **, *p* < 0.01). (**B**) Survival curve of WT and miR-10b KO mice with DSS treatment (*n* = 12/group). (**C**) Body weight changes were measured during the course of the experiment (*n* = 5/group; *, *p* < 0.05; **, *p* < 0.01). (**D**) Disease activity index score (*n* = 5/group; *, *p* < 0.05). (**E**) Typical colon images. (**F**) Quantification of colon length (*n* = 6/group; *, *p* < 0.05). (**G**) Histological images of colonic tissue (magnification, ×100; scale bar, 200 μm). (**H**) Histopathological score of colonic inflammation in mice with DSS treatment (*n* = 5/group; *, *p* < 0.05). (**I**) Epithelial permeability was determined using FITC-dextran assay (*n* = 5/group; **, *p* < 0.01). Data are representative of three repeated experiments.

**Figure 3 animals-13-01236-f003:**
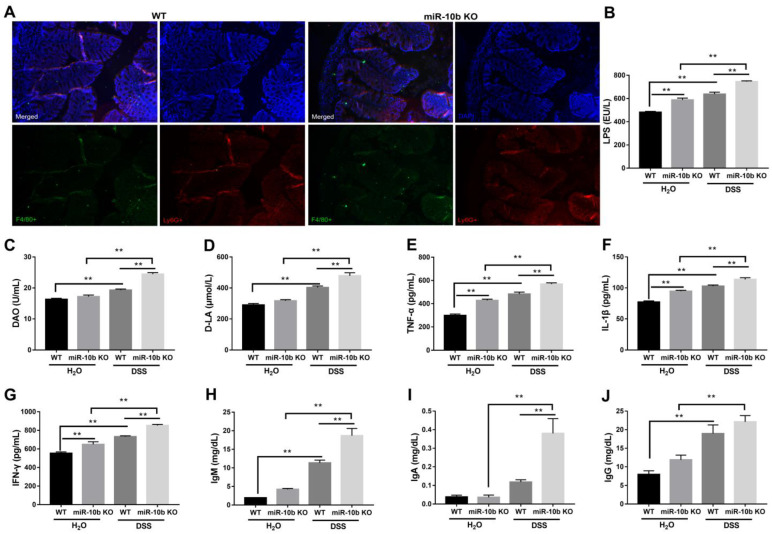
Exacerbated immune response in miR-10b KO mice following DSS treatment. (**A**) Inflammatory cell infiltration of the colonic mucosa with DSS treatment. Colon sections were stained with anti-F4/80 (green) for macrophages and anti-Ly6G (red) for neutrophils (magnification, ×200; scale bar, 500 μm). Serum concentrations of (**B**) LPS, (**C**) DAO, (**D**) D-LA, (**E**) TNF-α, (**F**) IL-1β, (**G**) IFN-γ, (**H**) IgM, (**I**) IgA, and (**J**) IgG were measured using ELISA techniques with commercially available kits (*n* = 5/group; **, *p* < 0.01). Data are representative of three repeated experiments.

**Figure 4 animals-13-01236-f004:**
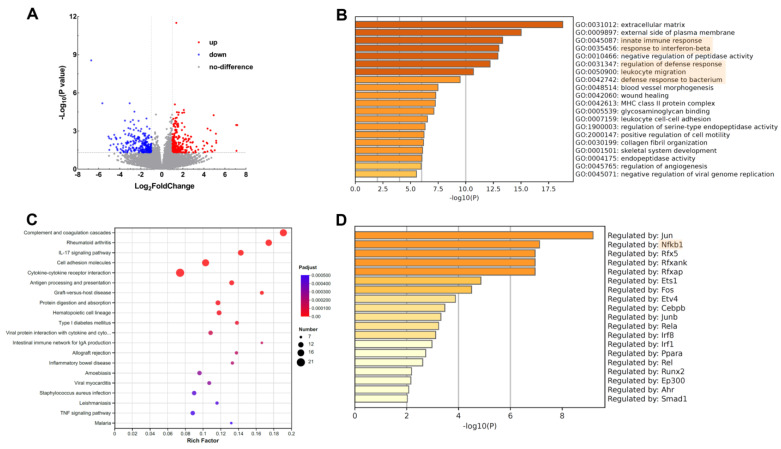
Bioinformatic analysis of the transcriptomes of inflamed colons after DSS treatment. (**A**) Differentially expressed genes between miR-10b KO and WT mice with DSS treatment were visualized using volcano plot. (**B**) The top 20 significantly enriched GO terms. (**C**) KEGG pathway enrichment analysis. The vertical axis represents various pathways, and the horizontal axis shows the corresponding rich factor. Higher rich factors suggest greater degrees of enrichment. Bubble size represents the number of genes enriched, while the color indicates statistical significance. (**D**) The top 20 core regulatory transcription factors were predicted using TRRUST enrichment analysis.

**Figure 5 animals-13-01236-f005:**
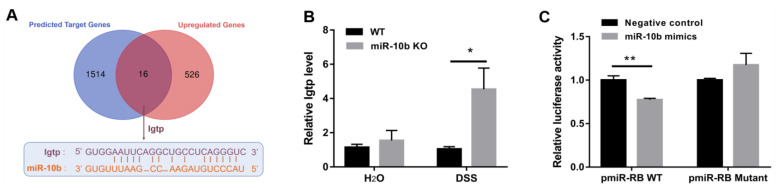
Identification of Igtp as a target of miR-10b. (**A**) The Venn diagram showing the overlap between the target genes of miR-10b predicted by StarBase and upregulated genes of miR-10b KO mice after DSS treatment. Igtp, one of the 16 overlapped genes, was considered as a potential key target gene of miR-10b. The schematic representation shows the sequence alignments of miR-10 with Igtp 3′ UTR. (**B**) The Igtp expression levels in the colon mucosa of mice were determined using qRT-PCR (*n* = 5/group; *, *p* < 0.05). (**C**) Luciferase activity was measured using the dual luciferase reporter assay system (*n* = 3/group; **, *p* < 0.01).

**Table 1 animals-13-01236-t001:** Primer sequences used for qRT-PCR analysis.

Genes	Primers	Sequences (5′-3′)
Igtp	Forward	CCGTGAACAAGTTCCTCAGGCT
	Reverse	GAGGTCTTGGTGTTCTCAGCCA
β-actin	Forward	GTGCTATGTTGCTCTAGACTTCG
	Reverse	ATGCCACAGGATTCCATACC
miR-10b	Forward	TACCCTGTAGAACCGAATTTGT
	Reverse	provided in the kit (Takara, China)

Abbreviations: Igtp = interferon gamma-induced GTPase; the universal reverse primer and the U6 primers were provided in the kit (Takara, Dalian, China).

## Data Availability

All data are available from the corresponding author upon reasonable request. Transcriptomes data are available from https://doi.org/10.6084/m9.figshare.21915900.

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
