# Peer review of "Investigation of the Expression Pattern and Functional Role of miR-10b in Intestinal Inflammation"

_animals, 2023, doi:10.3390/ani13071236_

Round 1
Reviewer 1 Report
Comments to authors:
This study provides mechanistic roles of miR-10b in intestinal inflammation. While the results clearly support the conclusion and extend our knowledge, there are some concerns which need to be addressed.
Major comments:
1. Was this study approved by relevant ethical society/committee?
2. The purpose of using murine model to understand the physiology in pig, in terms of functional role of miR 10b is never discussed. There seems to be an abrupt transition from pig to mice model.
3. Describe in detail how was the B-actin determined as a house-keeping gene? Did you also test for other genes as potential for normalization?
4. Ln 68-70: Provide reference
5. Ln 94: Was the expression of B-actin stable across samples? Please describe why/how B-actin was used was reference gene in this study?
6. Were the KO mice the same strain as the controls? If so, state it clearly.
7. What does each category of scores 0 - 4 for infiltration of inflammatory cells and cecal tissue damage look like. Please describe.
8. When was this study done? Why no to RNA-Seq? Microarray is quite outdated when we have now moved to single cell and spatial transcriptomics.
9. More details of methods of how disease activity index in Figure 2D was determined is necessary
10. Ln 199-201: … evidenced by more severe epithelial erosion, marked crypt damage, and increased 201 inflammatory cell infiltration. This result in not substantiated as the image (Figure 2G) is not label to show any histopathological abnormalities.
11. Ln 120: “Notably, increased macrophages numbers were found in the colonic lamina propria of 220 miR-10b KO mice”. Did you quantify this? If so, state the methods?
12. Ln 289-292: Discussion of the functional relevance of this statement is suggested.
13.
Minor comment:
Ln 48: Replace Whereas with while
Ln 63: Replace “was reported” to “has been reported to”
Ln 65: functional role in intestine?
Ln 65-68-In this study is redundant.
Ln 71: Delete: that is widely used and easy to induce
Ln 91: Please mention the samples used for RNA extraction.
Ln 94: Describe the acronym IGTP at its first use.
Ln 97: Describe the acronym U6 at its first use.
Ln 136: Describe the acronym F4/80 at its first use.
Ln 97: Describe the acronym U6 at its first use.
Ln 149-150: Please rephrase the DEG sentence. It is weirdly stated.
Figure 2B: The legend says **P but this is not visible in the actual figure.
Ln 221 and 224: Describe the acronyms at their first use.
Ln 265: Please describe miR-10b mimics in more detail.
Ln 276: Double check the significance **P.
Ln 301: Delete which.
Author Response
Major comments:
- Was this study approved by relevant ethical society/committee?
Response: We appreciate the reviewer’s positive evaluation of our work. This study was approved by ethical committee, “Institutional Review Board Statement: All experimental procedures were reviewed and approved by the Animal Care and Use Committee of Hunan Normal University, Hunan, China (2018-056).”, on page 12, lines 400-401.
- The purpose of using murine model to understand the physiology in pig, in terms of functional role of miR 10b is never discussed. There seems to be an abrupt transition from pig to mice model.
Response: We are very grateful for your careful work and thoughtful suggestions that have helped improve this paper substantially. We have discussed this issue as suggested, on page 2, lines 68-72.
- Describe in detail how was the B-actin determined as a house-keeping gene? Did you also test for other genes as potential for normalization?
Response: Thank you for your careful review. β-actin and glyceraldehyde-3-phosphate dehydrogenase (GAPDH) have been frequently considered as constitutive house-keeping genes for RT-PCR and used to normalize changes in specific gene expressions. We have used these two genes for normalization and our data demonstrated that the overall results were similar.
- Ln 68-70: Provide reference
Response: Thank you for your careful review. We have added a reference [15] as suggested, on page 2, lines 68-70.
- Ln 94: Was the expression of B-actin stable across samples? Please describe why/how B-actin was used was reference gene in this study?
Response: Thank you for your careful review. The expression of β-actin was stable across samples. β-actin and glyceraldehyde-3-phosphate dehydrogenase (GAPDH) have been frequently considered as constitutive house keeping genes for RT-PCR and used to normalize changes in specific gene expressions. We have used these two genes for normalization and our data demonstrated that the overall results were similar.
- Were the KO mice the same strain as the controls? If so, state it clearly.
Response: Thanks for your valuable advice. We have stated the background of KO mice as suggested, on page 3, line 115.
- What does each category of scores 0 - 4 for infiltration of inflammatory cells and cecal tissue damage look like. Please describe.
Response: Thank you very much for your careful review. We have described the scoring system in Supplementary Table S2.
- When was this study done? Why no to RNA-Seq? Microarray is quite outdated when we have now moved to single cell and spatial transcriptomics.
Response: We are very sorry for our mistake. We performed RNA-sequencing by the Illumina Hiseq platform in our study, and we have provided more details to bioinformatics analysis section, on pages 4-5, lines 159-177.
- More details of methods of how disease activity index in Figure 2D was determined is necessary
Response: We are grateful for the suggestion and we have described the scoring system in Supplementary Table S1.
- Ln 199-201: … evidenced by more severe epithelial erosion, marked crypt damage, and increased 201 inflammatory cell infiltration. This result in not substantiated as the image (Figure 2G) is not label to show any histopathological abnormalities.
Response: Thank you for your careful review. We have revised Figure 2 accordingly.
- Ln 120: “Notably, increased macrophages numbers were found in the colonic lamina propria of 220 miR-10b KO mice”. Did you quantify this? If so, state the methods?
Response: Thank you for your careful review. We have stated the methods on page 4, lines 148-150.
- Ln 289-292: Discussion of the functional relevance of this statement is suggested.
Response: Thank you for your insightful and constructive comments. We have reworded this statement on page 11, lines 314-317.
Minor comment:
Ln 48: Replace Whereas with while
Response: Thank you for your careful review. We have replaced “whereas” with “while” as suggested.
Ln 63: Replace “was reported” to “has been reported to”
Response: We have replaced “was reported” with “has been reported to” as suggested.
Ln 65: functional role in intestine?
Response: We have revised this sentence as suggested.
Ln 65-68-In this study is redundant.
Response: We have deleted “In this study” as suggested.
Ln 71: Delete: that is widely used and easy to induce
Response: We have deleted “that is widely used and easy to induce” according to your advice.
Ln 91: Please mention the samples used for RNA extraction.
Response: We have described the samples used for RNA extraction on page 3, lines 96, 101.
Ln 94: Describe the acronym IGTP at its first use.
Response: We have described the acronym IGTP as suggested.
Ln 97: Describe the acronym U6 at its first use.
Response: We have described the acronym U6 as suggested.
Ln 136: Describe the acronym F4/80 at its first use.
Response: We have described the acronym F4/80 as suggested.
Ln 149-150: Please rephrase the DEG sentence. It is weirdly stated.
Response: Thank you for your careful review. We have revised this part accordingly.
Figure 2B: The legend says **P but this is not visible in the actual figure.
Response: Thank you for your careful review. Survival curves of mice with DSS treatment were compared using Kaplan-Meier analysis and log-rank test. The P value indicated the overall survival differences between two groups.
Ln 221 and 224: Describe the acronyms at their first use.
Response: We have described the acronyms on page 4, lines 152-154.
Ln 265: Please describe miR-10b mimics in more detail.
Response: We have described miR-10b mimics on page 5, lines 185-186.
Ln 276: Double check the significance **P.
Response: Thank you for your careful review. We have revised P value accordingly.
Ln 301: Delete which.
Response: We have deleted “which” as suggested.

Reviewer 2 Report
Wu et al Investigated the expression pattern and functional role of miR-10b in intestinal inflammation in piglets. miRNAs play a vital role in maintaining the homeostasis of intestinal immune system and dysregulation of miRNAs expression has been linked to several inflammatory diseases. Therefore, the authors investigated the importance of miR-10b in immune regulation using a murine colitis model. The finding of the study is vital for management of swine specially during the weaning, which is associated with intestinal epithelium injury.
The authors presented the results well and the manuscript is well written. However, I have some concerns as indicated below.
Methodology section
This manuscript lacks essential information on library preparation for miRNA sequencing. To ensure reproducibility of the data please provide key information including kits and parameters used.
2.8 Bioinformatics analysis
This section lacks important information/key points regarding bioinformatics analysis of sequencing data.
What are the bioinformatics tools used in each step of the analysis to reach DEGs?
How is the quality control of the sequences done during the bioinformatics analysis of sequencing data?
Also mention the key steps regarding DEG analysis and what is the tool used for that step.
Author Response
What are the bioinformatics tools used in each step of the analysis to reach DEGs?
Response: We appreciate the reviewer’s positive evaluation of our work. We are also very grateful for the reviewer’s thoughtful comments. We have provided more details about the bioinformatics tools used as suggested, on pages 4, lines 167-169.
How is the quality control of the sequences done during the bioinformatics analysis of sequencing data?
Response: Thank you very much for your careful review. The methods for quality control have been summarized in Supplementary Table S3 and described on page 4, lines 160-167.
Also mention the key steps regarding DEG analysis and what is the tool used for that step.
Response: Thank you for your insightful and constructive comments. We have revised the bioinformatics analysis according to your valuable advice, on pages 4-5, lines 169-176.

Reviewer 3 Report
The manuscript entitled "Investigation of the expression pattern and functional role of miR-10b in intestinal inflammation" describes the rle of miR-10b in intestinal inflammation. In my opionion, the work is well written, the material and methods and results sections are complete, clear and adequately described.
Author Response
Response: We appreciate the reviewer’s positive evaluation of our work.

Round 2
Reviewer 1 Report
Thank you for addressing my comments and revising the manuscript. I have two minor comments below.
Ø Mention if the RNA-Seq was performed using single or paired end and how many base pairs.
Ø The scale bar for H2O miR-10b KO on Figure 2G is missing. Were all these images taken at same magnification?
Author Response
Ø Mention if the RNA-Seq was performed using single or paired end and how many base pairs.
Response: Thank you for your careful review. Paired-end sequencing (2 × 150 bp) was used in our study, and we have added this information on page 4, lines 159-160.
Ø The scale bar for H2O miR-10b KO on Figure 2G is missing. Were all these images taken at same magnification?
Response: Thank you very much for your careful review. All images were taken at same magnification. We have revised Figure 2G according to your valuable advice (magnification, ×100; scale bar, 200 μm).
